# A mixed methods study to evaluate participatory mapping for rural water safety planning in western Kenya

Joseph Okotto-Okotto[1], Weiyu Yu[2], Emmah Kwoba[3], Samuel M. Thumbi[3,4,5], Lorna Grace Okotto[6], Peggy Wanza[3], Diogo Trajano Gomes da Silva[7], Jim Wright[2]*

1 Victoria Institute for Research on Environment and Development (VIRED) International, Rabuor, Kisumu, Kenya, 2 School of Geography and Environmental Science, University of Southampton, Highfield, Southampton, United Kingdom, 3 Centre for Global Health Research, Kenya Medical Research Institute, Kisumu, Kenya, 4 Paul G Allen School for Global Animal Health, Washington State University, Pullman, WA, United States of America, 5 Institute of Tropical and Infectious Diseases, University of Nairobi, Nairobi, Kenya, 6 School of Spatial Planning and Natural Resource Management, Jaramogi Oginga Odinga University of Science and Technology, Bondo, Kenya, 7 School of Environment and Technology, University of Brighton, Brighton, United Kingdom

* J.A.Wright@soton.ac.uk

**Data Availability Statement:** Water source survey data are available from http://dx.doi.org/10.5255/UKDA-SN-853860, whilst participatory mapping data sets are available from http://dx.doi.org/10.

## Abstract

Water safety planning is an approach to ensure safe drinking-water access through comprehensive risk assessment and water supply management from catchment to consumer. However, its uptake remains low in rural areas. Participatory mapping, the process of map creation for resource management by local communities, has yet to be used for rural water safety planning. In this mixed methods study, to evaluate the validity of participatory mapping outputs for rural water safety planning and assess community understanding of water safety, 140 community members in Siaya County, Kenya, attended ten village-level participatory mapping sessions. They mapped drinking-water sources, ranked their safety and mapped potential contamination hazards. Findings were triangulated against a questionnaire survey of 234 households, conducted in parallel. In contrast to source type ranking for international monitoring, workshop participants ranked rainwater's safety above piped water and identified source types such as broken pipes not explicitly recorded in water source typologies often used for formal monitoring. Participatory mapping also highlighted the overlap between livestock grazing areas and household water sources. These findings were corroborated by the household survey and subsequent participatory meetings. However, comparison with household survey data suggested participatory mapping outputs omitted some water sources and landscape-scale contamination hazards, such as open defecation areas or flood-prone areas. In follow-up visits, participant groups ranked remediation of rainwater harvesting systems as the most acceptable intervention to address hazards. We conclude that participatory mapping can complement other established approaches to rural water safety planning by capturing informally managed source use and facilitating community engagement.

5255/UKDA-SN-853705. Questionnaire survey data are available from https://dx.doi.org/10.5255/UKDA-SN-854302.

**Funding:** JAW, DGS, JOO, SMT. UK Medical Research Council (https://mrc.ukri.org/) & Department for International Development (https://www.gov.uk/government/organisations/foreign-commonwealth-development-office). Grant Ref.: MR/P024920/1. The funders had no role in study design, data collection and analysis, decision to publish, or preparation of the manuscript.

**Competing interests:** The authors have declared that no competing interests exist.

## Introduction

Water safety planning is an approach to ensuring safe water access through comprehensive risk assessment and management at all water supply stages from catchment to consumer [1], promoted by the World Health Organization (WHO) [2]. Water safety planning entails the systematic identification of hazards between catchment and point-of-use within a water supply system and their management through identification of critical control points necessary for making that source safe for human consumption, documented via a Water Safety Plan (WSP). It was initially developed for urban utilities, but even in urban areas, its uptake has been slow globally [3]. Generally, Sub-Saharan Africa has lagged behind other world regions, with initial uptake in Uganda by 2011 [4], and subsequent more widespread uptake elsewhere in Africa including by Kenyan utilities [5].

Further challenges exist in promoting uptake of water safety planning in rural areas. Particularly in developing countries, limited resources and remoteness make the dissemination and uptake of such procedures challenging. These factors also inhibit water quality testing, an essential part of a WSP [6, 7]. The rural WSP workflow recognises that many rural water supply systems (e.g.: boreholes, protected wells and small-scale rainwater harvesting or piped systems) are managed by communities, sometimes via water user committees, rather than trained water sector professionals. Among community members, who have not received specialised professional training, perceptions of contamination hazards may differ from those of professionals, with consequent implications for hazard management [6]. Where community-managed supplies exist, water safety education that target communities requires different approaches from those targeted at water sector professionals [7, 8]. In rural areas, use of multiple water sources for different purposes and in different seasons is also very common, an idea encapsulated in the concept of multiple use water services [9], now being incorporated into rural participatory water planning [10]. This complex picture of multiple water source use also presents further challenges for rural water safety planning, since sources used sporadically or for specific purposes may be informally managed and omitted from water point inventories [11], despite their potential public health significance.

Responding to rural water safety planning challenges, WHO developed a simplified water safety planning workflow for small-scale supplies [2, 12], which involves six tasks: community engagement and team assembly; community supply description; identification of hazards and existing control measures; development and implementation of an improvement plan; monitoring of plan effectiveness; and documentation and review. A subsequent systematic review [8] highlighted the need for further simplification and community engagement in rural WSPs. Participatory methods used in rural WSPs have included group hazard elicitation, ranking and scoring techniques [5, 13], role-playing [14], and educational games [13]. Typically, WSP community engagement has focussed on formally managed supplies such as piped systems on boreholes, but not on household-managed sources such as wells or rainwater systems or informal sources such as surface water collection points. In Kenya, guidelines for water safety planning are targeted at utility companies rather than communities managing their own supplies and thus predominantly focussed on urban areas [15]. However, these guidelines do incorporate identification and mapping of non-piped supplies often used in rural areas such as boreholes and surface waters alongside piped systems.

Participatory mapping has also been used to support community engagement during the early stages of water planning [13] and to support water resources management without a specific focus on water safety [16]. The technique is particularly well suited to rural water safety planning in low resource settings because it does not require specialist equipment or skills, elicits community knowledge of both supply system and catchment hazards, and provides a

mechanism for raising awareness of water safety issues among consumers [2]. Participatory mapping describes a set of tools used to integrate human perceptions of the landscape with biophysical landscape data through the capture of local knowledge as a spatially referenced map layer, thereby enabling integration or comparison with scientific data also held as map layers [17]. The approach aims to empower local communities by capturing local knowledge and priorities. Participatory mapping has been used to understand the location and utilisation of natural resources [18], health mapping, education, crime prevention, mobility, and water and sanitation planning [19]. An early participatory mapping study [16] facilitated an action research partnership between Non-Governmental Organisations and local communities in Kenya and five other countries. Communities formed committees and began planning and mapping water resources through village walks. Participatory mapping has been used to map urban water and sanitation infra-structure to inform planning [20, 21], inform total community sanitation through mapping of open defecation areas [22], and explore the implications of future urban growth scenarios for water and wastewater service provision [23]. Templates have been developed for mapping rural water supplies during the system assessment phase of a WSP [12] and an Andean study engaged with youth co-researchers to understand and address the causes of water contamination in rural Colombia [24]. However, the approach has not yet been evaluated for rural water safety planning [8] through triangulation with other information sources.

In this study, we aim to evaluate the validity of participatory mapping outputs for rural water safety planning in western Kenya, focussing on the community supply description and hazard identification stages of safety planning. In doing so, the study also seeks to identify how communities perceive the safety of different water sources and the level of agreement between community and scientific understanding of water safety.

## Materials and methods

### Study site and population

Ethical approval was obtained from the Faculty of Social and Human Sciences, University of Southampton (references: 31554 and 48834; approval dates: 12/02/2018 and 23/04/2019) and the Kenya Medical Research Institute (reference: KEMRI/SERU/CGHR/091/3493, approval date: 17/10/2017). Informed written consent was obtained from participants. The study took place in ten villages near Lake Victoria in Siaya County, Kenya, which are among 33 villages participating in an ongoing Population-Based Animal Syndrome Surveillance (PBASS) study [25]. Although some households use piped water, rainwater and hand-dug wells, 29% of Siaya households used streams, rivers, dams and other surface waters as their main water sources in 2011 [26]. Boreholes and hand dug wells first developed by the Lake Basin Development Authority [27] and other Non-Governmental OrganiSations are managed through community water committees where still operational. Most households used pit latrines, but 16% lacked sanitation facilities. Smallholder agriculture is widespread, with 55% of households owning cattle [25].

### Study design, sampling and recruitment

A mixed methods study was implemented following a parallel convergent design to enable comparison of community knowledge with quantitative data concerning hazards and drinking-water sources [28]. The qualitative component consisted of two participatory mapping workshops held in each village. Participatory mapping outputs were evaluated through comparison with quantitative household questionnaire responses and a linked water source survey.

The PBASS study's network of community mobilisers provided a gender-balanced list of 30 to 40 long-term, literate adult residents, familiar with water and/or livestock management. Depending on village size, between 12 and 18 participants were randomly selected from these lists, giving a total of 165 participants. 185 participants were selected for the follow-up workshops via the same process, 60% of whom had not participated in initial workshops. The parallel household questionnaire survey was primarily designed to assess livestock-related risk factors for contamination of household stored water, but included questions concerning domestic water source use and contamination hazards relevant to participatory workshops. Independently of workshop participant selection, 234 adult survey participants were randomly selected from among households with children under 5 years already participating in the PBASS study.

## Participatory mapping workshops

Hard copy maps were prepared using ArcGIS 10.5 from 0.5 metre spatial resolution satellite imagery (georectified true colour composite WorldView 2 imagery, acquired in March 2013) for each village. Map extents were set to 1km beyond each village's boundary, resulting in ten coloured A1 maps with scales ranging from 1:4,344 to 1:7,645, reflecting participatory mapping preparation guidance [29]. Maps were printed onto durable cloth with a graticule to enable subsequent georeferencing.

In the initial workshops between 11[th] July and 17[th] October 2018, participants discussed and agreed ground rules, a water source typology and the nature, origins, and pathways of water contamination. Via group discussion, participants listed village water source types, ordering these from safest to most hazardous. Participants then listed potential hazards that could contaminate these sources in a similar process. Workshop participants then located water sources and hazards via the base-maps, with each village group nominating a smaller knowledgeable group from amongst them to locate hazards and water sources on base-maps. Small group participants then drew drinking-water sources and hazard locations onto transparencies affixed to image maps using agreed symbols and feature types (point or polygon).

Hard copy map overlays were scanned and georeferenced using ground control points via either spline or first order polynomial transformations, with mapped features then manually digitised. Root Mean Square Errors (RMSE) for the village-level transformations used varied from 0.855 to 1.756 metres. Alongside the flip-charts and map transparencies, anonymised minutes were taken of each meeting.

In subsequent meetings between 11[th] February and 9[th] March 2020, participants reviewed and corrected hardcopy maps, water source and hazard rankings from the earlier workshops through plenary discussion. The facilitators provided an overview of relevant potential interventions, including rainwater harvesting remediation [30], structures (e.g. fencing) to separate livestock and people at water sources [31], chlorine dispensers at water points [32], Point-of-Use (POU) household water treatment and safe storage [33], and related educational and hygiene awareness. Participants proposed and ranked interventions to reduce contamination, using the maps to prioritize specific water points for remediation. Given recommendations that communities retain ownership of participatory mapping outputs [19], hardcopy maps were presented to village leaders for use in water safety planning.

## Household and water source survey

Questionnaire interviews were conducted with 234 households between 12[th] March and 24[th] May 2018 and again with 230 of these households between 20[th] November 2018 and 18[th] February 2019 to capture the wet and dry seasons. Respondents were asked about seasonal uses of

water sources, sanitation, and livestock presence at water sources. To quantify omission of water points during participatory mapping, the survey team then visited each household's drinking-water source, surveying locations using a non-differential integrated GlobalSat BC-337 Compact Flash GPS receiver, attached to an iPAQ Hp Personal Digital Assistant (PDA) device. Since water's organoleptic properties influence how consumers perceive its safety [34], surveyors visually observed colour, cloudiness, and visible particles in all source water samples. Turbidity was measured *in situ* using a Hanna Instruments HI 93703 Portable Turbidity Meter, and electro-conductivity using a COND3110 handheld meter, both calibrated daily.

## Analysis

To evaluate internal consistency of participatory mapping outputs, following similar studies [35], we examined whether each group's mapping effort was consistent with its source safety or hazard importance rankings. We then evaluated the consistency of participatory mapping outputs with external evidence, drawing on the concepts of concurrent and convergent validity, developed for participatory mapping evaluations [36]. Concurrent validity refers to consistency in importance of concepts, whilst convergent validity refers to the spatial coincidence of mapped features. To assess concurrent validity, we compared each group's ranking of water source safety with a water 'ladder' used for monitoring of Sustainable Development Goal (SDG) 6 [37] using Spearman's rank correlation in Stata version 16 [38]. This 'ladder' differentiates 'improved' sources protected from contamination by nature of their design, from 'unimproved sources' lacking structural protection measures, with surface waters (e.g. rivers, ponds) forming the lowest tier. To further evaluate concurrent validity, we examined water source use (drinking, cooking, other domestic uses and livestock watering) by season reported via questionnaire. We assumed households implicitly considered sources for drinking safest, then cooking, and then other domestic uses. Village-level questionnaire responses concerning open defecation and human-livestock at drinking-water sources were also compared with hazards identified through participatory mapping.

To evaluate convergent validity, within ArcGIS 10.7, we identified source locations from the water point survey that lacked a corresponding water point in participatory mapping outputs within 115 metres. We chose this distance threshold by estimating overall positional error from GPS receivers, georeferencing, basemaps, manual digitising, and participants' drawings. We used Fisher's exact test to examine whether water point omission rates varied by source type. We also compared participants' piped water locations with a digital pipeline map from the service provider (Siaya-Bondo Water and Sanitation Company). We also calculated the percentage of water points by source type and village omitted from the first workshop but captured through participant review in the follow-up workshop. Finally, to evaluate organoleptic water properties' influence on perceived water safety, Spearman's rank coefficient was calculated for water safety rankings versus mean electro-conductivity, turbidity, and sample cloudiness and clarity.

## Results

### Participant characteristics

Of 165 participants invited to the first participatory workshops, 140 (85%) attended, with 63 (45%) and 34 (47%) being women in large groups and smaller groups undertaking detailed mapping respectively. Attendance remained high at subsequent feedback meetings, with 80% of invitees attending, except for one village (Wang'arot) where attendance fell.

Most household survey respondents were women aged under 40 years with responsibility for water management in the home (**Table 1**).

**Table 1. Questionnaire survey respondent characteristics by initial and follow-up visit.**

| Respondent characteristic | First visit (n; %) | Second visit (n; %) |
|---|---|---|
| Female | 197 (84.2%) | 189 (82.2%) |
| Person responsible for decisions on water handling, management and safety | 194 (82.9%) | 185 (80.4%) |
| 18–29 years | 75 (32.1%) | 79 (34.4%) |
| 30–39 years | 85 (36.3%) | 71 (30.9%) |
| 40–59 years | 57 (24.4%) | 60 (26.1%) |
| 60+ years | 17 (7.3%) | 20 (8.7%) |

## Community perception of water source safety and contamination hazards

When participants ranked source types by safety (Table 2), rainwater harvesting was considered safest by seven villages, springs safest by two villages and piped water safest by one village. On probing, many participants believed rainwater came directly from God (*"pi mogwedhi"* or "noble or blessed water") and therefore could not harm users even if consumed untreated. It was thus less frequently treated via chlorine or flocculation with alum than surface waters. Participants ranked rainwater highly because it cost nothing, apart from initial investments in storage containers or tanks; was free from salinity, unlike groundwater; and was free of the chlorine odour and taste of piped water. Only participants from Ong'ielo considered piped water safer than rainwater. The perceived safety of piped water was undermined by frequent interruptions. Pipelines that had burst, either accidentally or through deliberate vandalism to access free water, were identified as a source with intermediate safety. Whilst some springs were perceived as safe, in some cases, communities dug dry season watering ponds for livestock around springs, so livestock and people shared some springs, compromising their safety. Until further explanation was provided, participants struggled to distinguish boreholes from protected wells. Direct consumption of surface waters was rated the most hazardous in all villages, with groundwater sources often considered of intermediate safety.

Spearman's rank correlation coefficients between participants' source safety rankings and the JMP ladder rankings were significant in five villages (rho> = 0.80; p<0.05), marginally

**Table 2. Water source safety rankings in ten villages in Siaya County, Kenya, derived from participatory workshops.**

| | Improved sources | | | | Unimproved sources | | Surface water sources | | |
|---|---|---|---|---|---|---|---|---|---|
| | Piped water | Water kiosk | Bore-hole | Rainwater harvesting | Well | Spring | Streams / rivers | Water pan / pond / *put* | Burst pipe |
| Got Bondo | 2 | 3 | 5 | 1 | 4 | | 7 | 6 | |
| Kaminogedo (*) | 2 | 3 | | 1 | 5 | 6 | 8 | 7 | 4 |
| Lwak | 5 | 6 | 4 | 2 | 3 | 1 | 9 | 8 | 7 |
| Ndwara | 2 | | | 1 | | | 5 | 4 | 3 |
| Ong'ielo (*) | 1 | 2 | | 3 | | 5 | | 6 | 4 |
| Rambugu (*) | 2 | 3 | 4 | 1 | 5 | | | 6 | |
| Sangla (*) | | | | 1 | | | | 2 | |
| Siger | 2 | | | 1 | | | 5 | 4 | 3 |
| Sinogo | 6 | 5 | 3 | 2 | 4 | 1 | 9 | 8 | 7 |
| Wang'arot | 2 | 3 | 5 | 1 | 4 | 6 | 9 | 8 | 7 |
| Median Ranking | 2 | 3 | 4 | 1 | 4 | 5 | 8 | 6 | 4 |

(1 = safest, 9 = most hazardous

* case study villages for maps in Fig 1. A pan is a dammed water course and a *put* is a water-filled roadside excavation pit.)

insignificant in Ndwara, Siger and Sinogo (0.05<p<0.10), and insignificant in Lwak (rho = 0.56; p = 0.11).

Participants identified several non-landscape hazards (e.g. high iron from rusting pumps and roof catchments, dirty collection containers) that could not be mapped (S1 Fig) and several landscape hazards (e.g. flood events, open defecation areas) (S2 Fig) that they did not subsequently map. Participants did however map cattle grazing areas, bankside erosion, hazardous pit latrines and burst pipes in some villages (S3 Fig). Participant groups varied in their ability to identify hazards. In Sangla, participants listed five hazards, whereas in Ong'ielo, 15 hazards were listed.

Fig 1 shows four contrasting case study villages: Kaminogedo (comparatively water secure); Ong'ielo (where burst pipes form an important water source); Sangla (comparatively water insecure); and Rambugu (where participation in mapping activities by livestock herders was low). Participants put considerable effort into mapping rainwater harvesting systems. Rainwater harvesting systems and points for fetching surface water are widespread in all four villages. Some water sources included in rankings were not mapped and vice versa. In Lwak, for example, piped water was ranked but not mapped, whilst in Sangla, kiosks were mapped but not ranked.

In two villages, Kaminogedo and Ong'ielo, piped water forms an important supply system component. Pipeline distribution maps obtained from the water utility agreed with participants' locations of piped water points, kiosks, and smaller diameter pipelines in Ong'ielo, running southwards alongside the tarmac road. Despite piped water availability in both villages, participants mapped burst pipes and surface water points (e.g. streams, rivers) as drinking-water sources. Of the two villages without piped water, in Rambugu, boreholes and protected wells are available, whereas in Sangla, where salinity inhibits groundwater potential, other than a kiosk, only surface drinking-water sources were available.

Across all villages, participants mapped two hazardous pit latrines and 11 burst pipes as point hazards and delineated 447.6 Ha of grazing areas and 3.34 Ha of bankside erosion (17.4% and 0.1% of the study area respectively). Because cattle graze alongside the water courses that form village boundaries, people and cattle come into close proximity at water pans, springs and streams (Fig 1). In contrast, standpipes and wells points are further from cattle grazing areas.

## Consistency of participatory mapping outputs with questionnaire and water source survey

As reported through the questionnaire survey (Table 3), in the wet season, almost all households used rainwater for drinking, cooking, and other domestic purposes, preferring this even to piped water. Lacking rainwater in the dry season, many households reported using piped water, whilst others used wells, springs or boreholes and some relied on surface waters. Some households used rainwater, piped or groundwater for drinking only and not for other domestic purposes, apparently rationing its use. In contrast, few households used surface waters for drinking, suggesting it was used as a last resort. Some households supplemented surface waters for livestock with rainwater in the wet season and piped or well water in the dry season.

There were village-level discrepancies between the questionnaire survey and participatory mapping outputs. For example, many questionnaire respondents reported piped water use in Lwak, but standpipes were not mapped, whilst spring use was reported in Ndwara, but not ranked or mapped in the workshop. No households reported using burst pipes via the questionnaire survey in Kaminogedo, yet workshop participants there reported widespread burst pipe use following deliberate, illegal tampering with the pipeline, mapping six burst pipes.

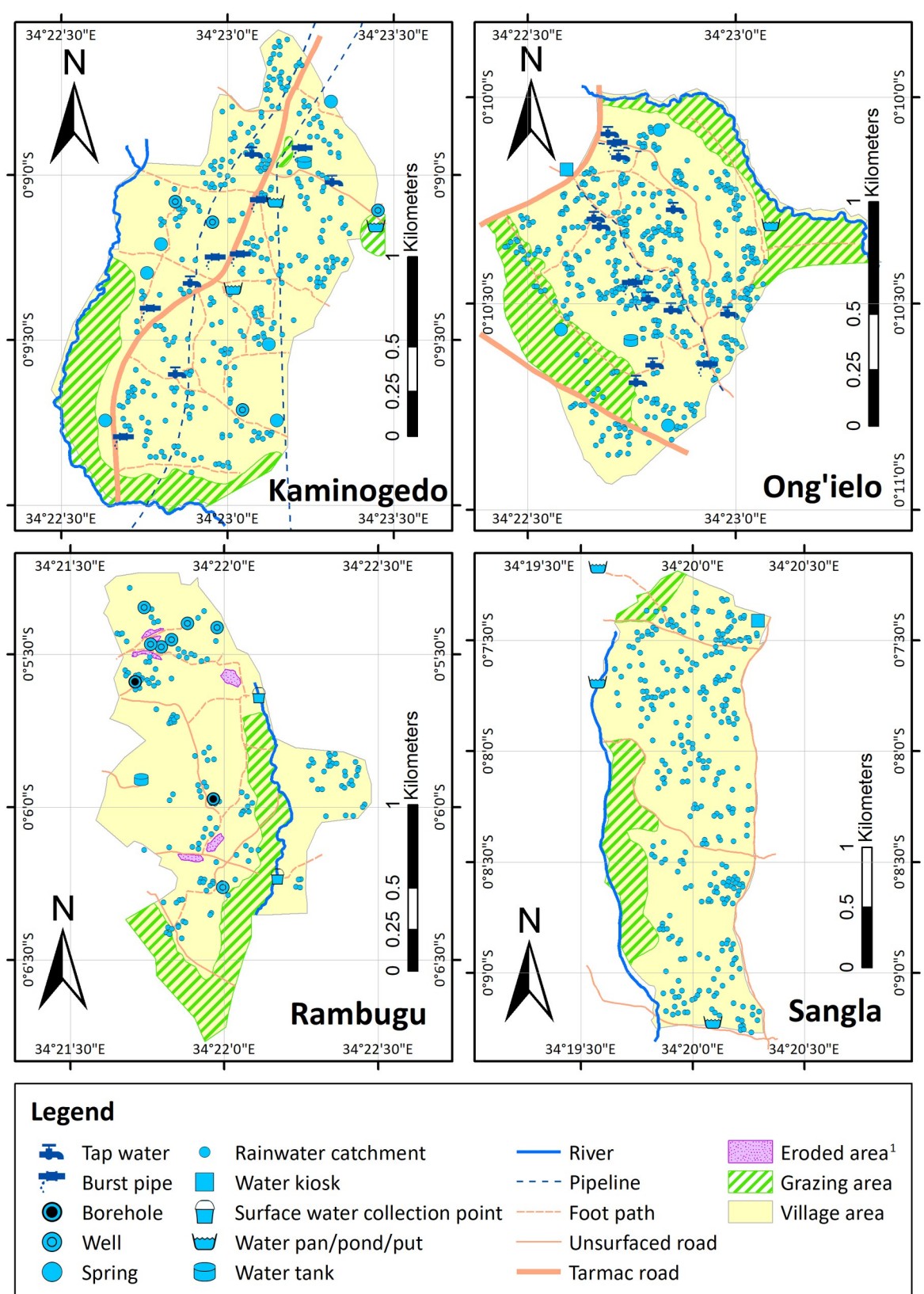

**Fig 1. Water sources and contamination hazards mapped by participants in four of the ten villages, overlaid on pipeline data from the local utility for Kaminogedo village.** ([1]Eroded areas as mapped by workshop participants).

**Table 3. Water source use, reported via two questionnaire survey visits (comprising 438 responses from 234 households).**

|  | Drinking | Cooking | Other domestic use | Livestock |
|---|---|---|---|---|
| **Wet Season** |  |  |  |  |
| Improved sources: |  |  |  |  |
| Piped water | 19 (4.1%) | 17 (3.7%) | 24 (5.2%) | 5 (2.2%) |
| Kiosk water | 0 (0.0%) | 1 (0.2%) | 1 (0.2%) | 0 (0.0%) |
| Boreholes | 1 (0.2%) | 1 (0.2%) | 0 (0.0%) | 0 (0.0%) |
| Rainwater | 451 (97.2%) | 448 (96.6%) | 437 (94.2%) | 44 (19.1%) |
| Predominantly unimproved sources: |  |  |  |  |
| Wells | 3 (0.6%) | 5 (1.1%) | 8 (1.7%) | 8 (3.5%) |
| Springs | 2 (0.4%) | 4 (0.9%) | 13 (2.8%) | 6 (2.6%) |
| Surface water: | 12 (2.6%) | 18 (3.9%) | 49 (10.6%) | 210 (91.3%) |
| Unclassified: |  |  |  |  |
| Burst pipes | 0 (0.0%) | 0 (0.0%) | 0 (0.0%) | 0 (0.0%) |
| **Dry Season** |  |  |  |  |
| Improved sources: |  |  |  |  |
| Piped water | 216 (46.6%) | 143 (30.8%) | 133 (28.7%) | 21 (9.1%) |
| Kiosk water | 1 (0.2%) | 1 (0.2%) | 4 (0.9%) | 1 (0.4%) |
| Boreholes | 12 (2.6%) | 9 (1.9%) | 7 (1.5%) | 1 (0.4%) |
| Rainwater | 37 (8.0%) | 7 (1.5%) | 3 (0.6%) | 1 (0.4%) |
| Predominantly unimproved sources: |  |  |  |  |
| Wells | 41 (8.8%) | 43 (9.3%) | 28 (6.0%) | 10 (4.3%) |
| Springs | 21 (4.5%) | 29 (6.3%) | 29 (6.3%) | 12 (5.2%) |
| Surface water | 172 (37.1%) | 271 (58.4%) | 293 (63.1%) | 222 (96.5%) |
| Unclassified: |  |  |  |  |
| Burst pipes | 0 (0.0%) | 3 (0.6%) | 4 (0.9%) | 2 (0.9%) |

Reinforcing hazards listed during workshops (**S1 Table**), at least three questionnaire respondents per village reported practicing open defecation, with 54 (23.2%) doing so overall. At least three households in each village also reported livestock and people sharing water sources during the dry season, with 82 (35.7%) doing so overall.

71.6% of water points used by questionnaire survey participants were recorded through participatory mapping (**Table 4**). According to Fisher's exact test, water point omission rates varied significantly by source type (p<0.001), with more private piped connections, surface water extraction points and burst pipes omitted than public standpipes or groundwater points.

*In situ* testing showed that piped or kiosk water, borehole water and rainwater were seldom coloured, cloudy or with visible particles, and had low turbidity unlike other source types (see **S2 Table**). Borehole water had the highest electro-conductivity, which would produce an objectionable salty taste for consumers of some borehole water [39]. Per source type, median water safety rankings from participatory mapping were positively correlated with low turbidity and negatively with samples coloured, cloudy or with visible particles (n = 8; rho = 0.80; p = 0.017; rho = 0.72; p = 0.046 respectively).

## Community follow-up workshops

During follow-up meetings, participants identified and mapped 13 new springs, 12 water pans, 11 wells, six river or lake water collection points, a standpipe and a burst pipe not captured through earlier participatory mapping (see **S2 Table**). Excluding rainwater harvesting points,

**Table 4. Proportion of water points in source survey captured (i.e. with a corresponding point of the same type within 115 metres) through participatory mapping, by source type.**

| Source type | % captured by participatory mapping | Number of water points surveyed |
|---|---|---|
| Improved sources: | | |
| Piped water: private connections | 38.1% | 21 |
| Piped water: public standpipes | 66.7% | 3 |
| Kiosk | 100% | 1 |
| Borehole | 100% | 3 |
| Rainwater | 97.2% | 36 |
| Predominantly unimproved sources: | | |
| Wells | 83.3% | 12 |
| Springs | 100% | 1 |
| Surface waters: | | |
| Stream / river water collection point | 33.3% | 6 |
| Pond/put/pan/lake | 60% | 10 |
| Unclassified source types: | | |
| Burst pipes | 0% | 2 |
| Total | 71.6% | 95 |

this suggested the initial workshops omitted 28% of water points, including 67% of surface water collection points, 50% of springs, 32% of water pans, 26% of wells, 8% of burst pipes, and 4% of standpipes. Village-level omission rates varied from 47% in Siger to 0% in Got Bondo.

In identifying potential interventions to address contamination (**S3 Table**), participants ranked remediation (e.g. gravel filter installation) or upgrading of rainwater harvesting systems highest, then well and water pan remediation measures (e.g. fencing to prevent livestock entry, provision of separate cattle troughs) lower. In some villages, participants also proposed borehole installation and proposed woodlot development to support boiling of water, because of fuelwood shortages and woodlots' perceived environmental benefits. Participants also identified those interventions requiring external support and used the maps generated previously to propose specific locations for highly ranked interventions.

## Discussion

Participatory mapping highlighted the importance of informally managed rainwater and surface water sources to the community. In the wet season, households consumed harvested rainwater in preference to piped water, implying that rainwater safety held greater public health significance than piped water. In the dry season, many households relied on surface water sources, sharing these with livestock, posing obvious risks for water-borne disease transmission. Many WSPs focus solely on formally managed water supplies (boreholes and piped water) lacking a participatory component and could thus overlook the importance of informal source types.

Community mapping and ranking of water sources is largely consistent with scientific understanding of water safety, suggesting high concurrent validity [36]. In ranking sources for water safety, the community recognised the risks of consuming surface waters, in keeping with scientific knowledge [40] and the JMP's 'ladder' used for international monitoring of SDG 6 [37]. However, in contrast to the JMP's 'ladder', rainwater was ranked above piped water. Thus, the hierarchy of water sources used for international monitoring of SDG 6 (i.e. the 'ladder') differs from the communities' ranking of different source types. In follow-up workshops,

the communities generally ranked rainwater harvesting remediation highest as an intervention, even though rainwater is placed on an intermediate rung in the 'ladder' used for international monitoring. High consumer regard for rainwater has also been observed via smart meters in Kenya [41] and in countries such as Fiji [6]. Community members appear to implicitly value rainwater's cheapness and supply continuity, alongside the explicit ranking criterion, its safety. Rainwater had low electro-conductivity, turbidity, and visible contamination signs, suggesting households considered it safe because of its organoleptic properties [34]. Despite its perceived safety, harvested rainwater is at risk of post-collection contamination through handling and storage [42]. Rainwater collected from rooftops may be contaminated by bird droppings, small mammal faeces and organic decomposition, with thermotolerant coliform contamination reported in several studies, particularly following rainfall [30]. Pathogens (e.g. giardia, campylobacter, and salmonella) have also been detected in rainwater [43], indicating public health risks.

The community's water source typology included some types not present in the core questions for international monitoring [44], implemented globally via household surveys such as Demographic and Health Surveys. Firstly, two communities reported use of broken pipes to access water, a category not in the standard classification. Secondly, participants reported adapting sources to enhance water security, such as impounding springs to facilitate easier access by people and livestock, forming a dam-spring 'hybrid'. Both 'hybrid' sources and (possibly deliberately) broken pipes reflect known 'exit' strategies for coping with water insecurity [45]. Sources reflecting household adaptations to water insecurity thus appear inadequately captured in quantitative surveys.

In terms of convergent validity, comparison with questionnaire survey data (**Table 4**) suggested that participatory mapping outputs frequently omitted private piped connections, surface water extraction points, and pipe breakages. Illegality of some piped breakages may have inhibited open discussion of these sources [46], whilst many private connections were likely unknown to the wider community. However, despite concerns over the validity of workshop-based participatory mapping outputs [47], rainwater systems, public standpipes and groundwater sources were well captured. Follow-up workshop participants mapped additional groundwater sources but failed to capture omitted private standpipe connections. Otherwise, omission patterns were similar to the questionnaire survey. Workshop participants struggled to distinguish boreholes from mechanised hand-dug wells, which could result in misclassification of self-reported water source types in household survey data sets, such as those used for international monitoring. Classification ambiguity is widespread in spatial databases [48], and so may affect other databases such as water point mapping data sets [11].

Comparison with questionnaire responses suggested workshop participants were able to map some, but not all, contamination hazards present in villages. Both the questionnaire survey and participatory workshops highlighted livestock and people sharing surface water sources as a concern alongside human open defecation. Participants successfully mapped some landscape-scale hazards subsequently, particularly cattle grazing areas. However, diffuse source- or household-scale hazards, such as rubbish pits, rusting pumps and unsanitary water collection vessels could not be mapped at landscape scale. Participants identified open defecation as a landscape-scale hazard, but were unable to map such areas. Furthermore, the number of hazards identified varied between villages, suggesting variable understanding of contamination hazards across participant groups. As with water sources, participatory mapping outputs thus only partially captured the landscape-scale contamination hazards present. It also suggests a need for follow-up health education in some communities.

Initial and follow-up workshops highlighted the importance of household and informally managed sources for the community. Initial workshops identified surface water collection

points shared between livestock and people as particularly hazardous. Furthermore, many of the solutions proposed by communities (**S3 Table**) related either to informally managed sources, such as separate troughs for cattle at surface water collection points, or to household sources, such as rainwater harvesting systems. It has previously been noted that complex water systems require complex water safety plans and that pre-existing community organisational structures make safety plan establishment more straightforward [49]. Whilst the supply systems used in the ten villages here are not technically complex, the household pattern of multiple water source use is complex. Multiple water source use has been reported in many other rural areas [9]. Furthermore, pre-existing community organisational structures typically relate to formal water sources such as boreholes, but are often lacking for informal sources such as surface water collection points. Thus, despite the need for simplification of rural water safety planning procedures [13], the mix of formal and informal sources used makes rural water safety planning complex and challenging.

Several organisational aspects of participatory mapping are likely to have influenced its outcomes. In this participatory mapping exercise, livestock herders were explicitly recruited as participants and the informed consenting process explained the project's purpose, investigation of livestock-related water contamination. Since discussions followed a different trajectory in the community where livestock herders were under-recruited, this may have resulted in participant groups mapping grazing areas as hazards, but potentially omitting hazards such as open defecation areas. Whilst participant characteristics can profoundly affect focus group discussions [50], when participants elected smaller groups to undertake mapping, women remained well represented in those elected.

Whilst we rapidly mobilised village communities for our study, community mobilisation to support rural WSPs could be more difficult elsewhere. Our study area falls under a Health and Demographic Surveillance System (HDSS), which has previously evaluated HIV services, integrated malaria control, and rotavirus vaccine [51]. Following subsequent successful take-up of some resultant interventions, a relationship of trust has developed between residents and researchers, increasing engagement in research. The PBASS village reporter network facilitated participant recruitment, but this community mobilisation resource would typically be lacking elsewhere. The participatory mapping facilitator in our study was also highly experienced, having previously led similar activities elsewhere [23, 52], and resources were available for preparing, printing, and digitising hardcopy maps. Thus, a lack of similarly qualified staff, resources, and community trust elsewhere [53] could inhibit uptake of participatory mapping beyond our study site.

We have used participatory mapping for just three WSP stages [2] here: community engagement, supply description and hazard identification and stakeholder feedback. There would be scope to expand its use to support the subsequent WSP implementation and monitoring stages, which we intend to do. Since our study suggests communities have a unique, spatial perspective on water sources and hazards, this could be used to inform the design of critical control points for managing water safety. A WSP that leverages the high community regard for rainwater could for example be worth exploring. Within the system evaluation stage, our protocol could be further refined through integration with mapping templates for rural piped system WSPs [12]. Map symbols for treatment and storage facilities in these templates could be expanded to enable communities to document control measures at informal sources (e.g. fencing around boreholes or wells, or rainwater tanks fitted with gravel filters). It would also be straightforward to incorporate hazard ranking into future participatory mapping. Given that a Fijian study [6] highlighted minimal household knowledge of post-collection contamination risks during rainwater handling and storage, participatory mapping should form just one component for landscape-scale catchment hazard assessment within a

comprehensive community-based WSP. It should not be adopted whilst ignoring the well-documented risk of post-collection water recontamination [42].

## Conclusions

Our use of participatory mapping highlighted the public health importance of informally managed surface and rainwater sources, which would be overlooked were a WSP to cover only formally managed piped water and boreholes. As confirmed by a parallel questionnaire survey, communities frequently consumed rainwater without treatment, in preference to piped water in the wet season, valuing rainwater highly. Many households also shared surface water sources with cattle during the dry season. However, comparison with the questionnaire survey suggested that participatory mapping outputs omitted some water sources and landscape-scale contamination hazards. For example, standpipes remained unmapped in some villages and no participant groups mapped hazards such as flood-prone or open defecation areas. Some groups also struggled to identify many water contamination hazards, suggesting a need for follow-up health education. This suggests participatory mapping can form a valuable complement to rural water safety planning in capturing informal source use, alongside established approaches such as catchment surveys and formal supply system documentation. The technique could be used more widely to engage with rural communities during the system assessment phase of water safety planning. Future research could explore its value for community engagement in the later implementation and monitoring WSP stages. Our study identified informal surface water collection points as particularly hazardous through participatory mapping, and communities viewed household-managed rainwater harvesting systems as a priority for remediation. Therefore, future research should also assess how far the rural water safety planning approach can be applied to informal and household-managed sources alongside the formal sources that have largely been the focus of water safety planning to date.

## Supporting information

**S1 Fig. Numbers of unmappable contamination hazards identified by workshop participants in ten villages.**
(TIF)

**S2 Fig. Numbers of unmapped contamination hazards identified by workshop participants in ten villages.**
(TIF)

**S3 Fig. Numbers of mapped contamination hazards identified by workshop participants in ten villages.**
(TIF)

**S1 Table. Counts of new water points mapped in second workshop by village and source type.**
(XLSX)

**S2 Table. Physico-chemical and organoleptic water properties from a water source survey in ten villages in Siaya County.**
(DOCX)

**S3 Table. Rankings of community interventions proposed by participants during follow-up meetings.**
(DOCX)

**S1 Protocol. Household Survey Questionnaire.**
(DOCX)

**S2 Protocol. Participatory mapping topic guide.**
(DOCX)

## Acknowledgments

The authors thank the Digital Scholarship Team at the University of Southampton's Hartley Library for large map scanning support.

## Author Contributions

**Conceptualization:** Joseph Okotto-Okotto, Samuel M. Thumbi, Lorna Grace Okotto, Diogo Trajano Gomes da Silva, Jim Wright.

**Data curation:** Weiyu Yu, Peggy Wanza.

**Formal analysis:** Weiyu Yu, Jim Wright.

**Funding acquisition:** Joseph Okotto-Okotto, Samuel M. Thumbi, Jim Wright.

**Investigation:** Joseph Okotto-Okotto, Emmah Kwoba, Diogo Trajano Gomes da Silva.

**Methodology:** Joseph Okotto-Okotto, Emmah Kwoba, Lorna Grace Okotto, Diogo Trajano Gomes da Silva.

**Project administration:** Joseph Okotto-Okotto, Samuel M. Thumbi, Diogo Trajano Gomes da Silva, Jim Wright.

**Software:** Peggy Wanza.

**Supervision:** Joseph Okotto-Okotto, Emmah Kwoba, Samuel M. Thumbi.

**Visualization:** Weiyu Yu.

**Writing – original draft:** Joseph Okotto-Okotto, Jim Wright.

**Writing – review & editing:** Joseph Okotto-Okotto, Weiyu Yu, Emmah Kwoba, Samuel M. Thumbi, Lorna Grace Okotto, Peggy Wanza, Diogo Trajano Gomes da Silva, Jim Wright.

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
