## [Decision Letter · Decision Letter 0]

15 Dec 2020

PONE-D-20-28371

A mixed methods study to evaluate participatory mapping for rural water safety planning in western Kenya

PLOS ONE

Dear Dr. Wright,

Thank you for submitting your manuscript to PLOS ONE. After careful consideration, we feel that it has merit but does not fully meet PLOS ONE’s publication criteria as it currently stands. Therefore, we invite you to submit a revised version of the manuscript that addresses the points raised during the review process.

The manuscript “A mixed methods study to evaluate participatory mapping for rural water safety planning in western Kenya” was reviewed by two independent reviewers, both of who commented positively on the approach developed in the study. However, both also highlighted several areas of concern related to placing the study within the international context as well as the local context. In this sense, it would be helpful to better highlight how the study relates to the ongoing use of participatory approaches (mapping, citizen science) for water management, one reviewer provided a number of related publications. For the latter, more attention should be dedicated to the next steps in the local context related to rural water safety planning and related research. I would also suggest that some attention should be placed, positioning the study in relation to UN SDGs, as this would improve the visibility of the manuscript, once published

We look forward to receiving your revised manuscript.

Kind regards,

Steven Arthur Loiselle

Academic Editor

PLOS ONE

Journal Requirements:

2.) Please ensure you have provided a copy of all questionnaires used in your study as supporting information files.

3.) We note that Figure 2 in your submission contains map images which may be copyrighted. All PLOS content is published under the Creative Commons Attribution License (CC BY 4.0), which means that the manuscript, images, and Supporting Information files will be freely available online, and any third party is permitted to access, download, copy, distribute, and use these materials in any way, even commercially, with proper attribution. For these reasons, we cannot publish previously copyrighted maps or satellite images created using proprietary data, such as Google software (Google Maps, Street View, and Earth). For more information, see our copyright guidelines: http://journals.plos.org/plosone/s/licenses-and-copyright.

You may seek permission from the original copyright holder of Figure 2 to publish the content specifically under the CC BY 4.0 license. 

If you are unable to obtain permission from the original copyright holder to publish these figures under the CC BY 4.0 license or if the copyright holder’s requirements are incompatible with the CC BY 4.0 license, please either i) remove the figure or ii) supply a replacement figure that complies with the CC BY 4.0 license. Please check copyright information on all replacement figures and update the figure caption with source information. If applicable, please specify in the figure caption text when a figure is similar but not identical to the original image and is therefore for illustrative purposes only.

Reviewers' comments:

Reviewer's Responses to Questions

**Comments to the Author**

1. Is the manuscript technically sound, and do the data support the conclusions?

Reviewer #1: Yes

Reviewer #2: Yes

2. Has the statistical analysis been performed appropriately and rigorously? 

Reviewer #1: Yes

Reviewer #2: Yes

3. Have the authors made all data underlying the findings in their manuscript fully available?

Reviewer #1: Yes

Reviewer #2: Yes

4. Is the manuscript presented in an intelligible fashion and written in standard English?

Reviewer #1: Yes

Reviewer #2: Yes

5. Review Comments to the Author

Reviewer #1: Manuscript Number: PONE-D-20-28371

Title: A mixed methods study to evaluate participatory mapping for rural water safety planning in western Kenya

This manuscript introduces a mixed methods approach for mapping water safety planning in Kenya. The premise of the manuscript is that it introduces a “participatory mapping” to rural water safety planning –a method that has never been applied to the empirical study or practice of rural water safety planning.

Generally speaking, this manuscript presents an interesting methodology and case study. It is well written and is relevant; however, it warrants some revisions to render it publishable in PLOS One. In order to be constructive, I structure my impressions of the article around two interrelated aspects: 1) the theoretical framing of the manuscript (i.e., the literature review); and 2) the empirical framing of the manuscript (i.e., research methods and analysis).

1) The theoretical framing of the manuscript (i.e., the literature review)

Basically, this component is missing from this manuscript. While the introduction is well written, it does not situate the manuscript’s topic “planning for rural water safety” within the larger body of debates on the subject matter. What is the status of rural water safety planning generally speaking, in Africa, and in Kenya particularly? what are the primary concerns with regards to water planning in rural Kenya? how has participatory planning for rural water safety achieved in general, in Africa, and in Kenya? and why is a participatory planning for rural water safety important (in comparison to other methods/approaches)? Including this discussion will render manuscript relevant to a larger audience of readers and will render some components of it transferable (e.g., the methods) and generalizable (e.g., the findings).

Similarly, and because the manuscript is built on the claim that “participatory mapping” has never been applied to the empirical study or practice of rural water safety planning, the authors should introduce what exactly is “participatory mapping”? How has it been applied in the past, for which aspects of rural planning, and in which rural contexts within Africa and/or Kenya? And how is this method transferable to mapping rural water safety in particular? And, building on the previous question, why is this method optimal? Again, through addressing these questions, the readers would be provided with a context for this study.

Some references that should be consulted in this process include, but are not limited to:

I. On participatory mapping in Africa (note that some actually pertain to rural water planning):

a) Fagerholm, N., & Käyhkö, N. (2009). Participatory mapping and geographical patterns of the social landscape values of rural communities in Zanzibar, Tanzania. Fennia - International Journal of Geography, 187(1), 43-60. Retrieved from https://fennia.journal.fi/article/view/3703

b) Jiri Panek (2015) How participatory mapping can drive community empowerment – a case study of Koffiekraal, South Africa, South African Geographical Journal, 97:1, 18-30, DOI: 10.1080/03736245.2014.924866

c) Lammerink, Marc P. “Community Managed Rural Water Supply: Experiences from Participatory Action Research in Kenya, Cameroon, Nepal, Pakistan, Guatemala and Colombia.” Community Development Journal, vol. 33, no. 4, 1998, pp. 342–352. JSTOR, www.jstor.org/stable/44258804.

II. On participatory mapping in general and/or in contexts outside Africa:

d) Chambers, R. (2006), Participatory Mapping and Geographic Information Systems: Whose Map? Who is Empowered and Who Disempowered? Who Gains and Who Loses?. The Electronic Journal of Information Systems in Developing Countries, 25: 1-11. https://doi.org/10.1002/j.1681-4835.2006.tb00163.x

e) M. Anwar Hossen (2016) Participatory mapping for community empowerment, Asian Geographer, 33:2, 97-113, DOI: 10.1080/10225706.2016.1237370

f) C.E. Roa García, S. Brown (2009). Assessing water use and quality through youth participatory research in a rural Andean watershed, Journal of Environmental Management, Volume 90, Issue 10, Pages 3040-3047

2) The research methods and analyses:

This discussion that spreads over sections 2 and 3 is well-written and reflects a sound and strong methodology. However, it is extremely lengthy and overly detailed –to the extent that the bigger picture behind this empirical study is diluted. I recommend condensing these sections by at least a third each, if not more, and elaborating on the bigger picture and the relevance of the study’s findings to planning for rural water safety within Kenya specifically, but also, how does this study (whether its methods and/or findings) inform planning for rural water safety in Africa and beyond. In other words, the manuscript should highlight this study’s contribution and insights for rural Kenya and beyond.

Reviewer #2: This is a good study that empowers local communities to engage in water security. The ability to triangulate the outcomes between the surveys and the participatory mapping was critical to demonstrating the effectiveness of their methods. However, the overall article ends weakly. In particular the last two sentences let the reader down. The next steps and recommendations for future research should be clearly articulated at the end of the conclusion section. Some of this is hinted at in the discussion, finish strongly.

6. PLOS authors have the option to publish the peer review history of their article (what does this mean?). If published, this will include your full peer review and any attached files.

Reviewer #1: No

Reviewer #2: No

---

## [Author Response · Author response to Decision Letter 0]

1 Feb 2021

Editorial comments:

The editor recommended that we strengthen the linkages between our work and the UN SDGs. With this in mind, we have clarified in our methods section that the WHO/UNICEF Joint Monitoring Programme's 'ladder' of water sources, which we use in our analysis, forms the basis for monitoring of SDG 6, Target 6.1. We also now reflect in paragraph 2 of our revised Discussion on the differences between this 'ladder' used for SDG 6 monitoring and how local communities rank and prioritise different water source types in our study.

Reviewer #1: 

This manuscript introduces a mixed methods approach for mapping water safety planning in Kenya. The premise of the manuscript is that it introduces a “participatory mapping” to rural water safety planning –a method that has never been applied to the empirical study or practice of rural water safety planning.

Generally speaking, this manuscript presents an interesting methodology and case study. It is well written and is relevant; however, it warrants some revisions to render it publishable in PLOS One. In order to be constructive, I structure my impressions of the article around two interrelated aspects: 1) the theoretical framing of the manuscript (i.e., the literature review); and 2) the empirical framing of the manuscript (i.e., research methods and analysis).

1) The theoretical framing of the manuscript (i.e., the literature review)

COMMENT: Basically, this component is missing from this manuscript. While the introduction is well written, it does not situate the manuscript’s topic “planning for rural water safety” within the larger body of debates on the subject matter. What is the status of rural water safety planning generally speaking, in Africa, and in Kenya particularly? what are the primary concerns with regards to water planning in rural Kenya? how has participatory planning for rural water safety achieved in general, in Africa, and in Kenya? and why is a participatory planning for rural water safety important (in comparison to other methods/approaches)? Including this discussion will render manuscript relevant to a larger audience of readers and will render some components of it transferable (e.g., the methods) and generalizable (e.g., the findings).

RESPONSE: We accept that whilst we had explained water safety planning and the challenges for rural areas in our Introduction, we had not summarised its status and uptake in different settings. We now do so in a revised paragraph 1, drawing on relevant references including some helpfully suggested by the reviewer (one of which we had cited in our original manuscript though not in the Introduction). To strengthen the rationale for exploring participatory mapping in relation to rural water safety planning, we have broken paragraph 2 of our original Introduction out into two new paragraphs (now forming paragraphs 2 and 3 of the revised Introduction). Paragraph 2 sets out the challenges of implementing water safety planning in rural areas of developing countries. Via a deepened literature synthesis, Paragraph 3 reports on the ‘state of the art’, summarising efforts to simplify water safety planning in such settings, and participatory methods previously applied for engaging with community groups such as water user committees.

COMMENT: Similarly, and because the manuscript is built on the claim that “participatory mapping” has never been applied to the empirical study or practice of rural water safety planning, the authors should introduce what exactly is “participatory mapping”? How has it been applied in the past, for which aspects of rural planning, and in which rural contexts within Africa and/or Kenya? And how is this method transferable to mapping rural water safety in particular? And, building on the previous question, why is this method optimal? Again, through addressing these questions, the readers would be provided with a context for this study.

Some references that should be consulted in this process include, but are not limited to:

I. On participatory mapping in Africa (note that some actually pertain to rural water planning):

a) Fagerholm, N., & Käyhkö, N. (2009). Participatory mapping and geographical patterns of the social landscape values of rural communities in Zanzibar, Tanzania. Fennia - International Journal of Geography, 187(1), 43-60. Retrieved from https://fennia.journal.fi/article/view/3703

b) Jiri Panek (2015) How participatory mapping can drive community empowerment – a case study of Koffiekraal, South Africa, South African Geographical Journal, 97:1, 18-30, DOI: 10.1080/03736245.2014.924866

c) Lammerink, Marc P. “Community Managed Rural Water Supply: Experiences from Participatory Action Research in Kenya, Cameroon, Nepal, Pakistan, Guatemala and Colombia.” Community Development Journal, vol. 33, no. 4, 1998, pp. 342–352. JSTOR, www.jstor.org/stable/44258804.

II. On participatory mapping in general and/or in contexts outside Africa:

d) Chambers, R. (2006), Participatory Mapping and Geographic Information Systems: Whose Map? Who is Empowered and Who Disempowered? Who Gains and Who Loses?. The Electronic Journal of Information Systems in Developing Countries, 25: 1-11. https://doi.org/10.1002/j.1681-4835.2006.tb00163.x

e) M. Anwar Hossen (2016) Participatory mapping for community empowerment, Asian Geographer, 33:2, 97-113, DOI: 10.1080/10225706.2016.1237370

f) C.E. Roa García, S. Brown (2009). Assessing water use and quality through youth participatory research in a rural Andean watershed, Journal of Environmental Management, Volume 90, Issue 10, Pages 3040-3047

RESPONSE: We agree we should have more tightly defined the manuscript’s fit to prior knowledge via a more extensive and deeper literature review. We also thank the reviewer for taking the time to highlight specific relevant references for us. Paragraph 3 of the Introduction of our original manuscript provided a brief summary of participatory mapping and its applications. In the light of the reviewer’s comments and request that we deepen the literature review and more clearly explain the specific contribution of the manuscript, we have expanded this paragraph to incorporate some of the references above, strengthening the case for its application to rural water safety planning in doing so. We also clarify that we are not proposing that participatory mapping is the optimal method for rural water safety planning. Rather, we argue it is one of several relevant participatory techniques with potential for water safety planning. As such, we seek to provide evidence on its usefulness or otherwise through triangulation with other sources of information.

2) The research methods and analyses:

This discussion that spreads over sections 2 and 3 is well-written and reflects a sound and strong methodology. However, it is extremely lengthy and overly detailed –to the extent that the bigger picture behind this empirical study is diluted. I recommend condensing these sections by at least a third each, if not more, and elaborating on the bigger picture and the relevance of the study’s findings to planning for rural water safety within Kenya specifically, but also, how does this study (whether its methods and/or findings) inform planning for rural water safety in Africa and beyond. In other words, the manuscript should highlight this study’s contribution and insights for rural Kenya and beyond.

RESPONSE: As suggested, we have shortened Section 2 (Methods) by about a third for brevity, reducing methodological detail in doing so. To shorten Section 3 (Results), we have deleted Tables 1 and 4, summarising key statistics in the text instead. We also moved Figure 1 and Table 7 to Supplemental Materials, thereby nearly halving the number of illustrations. In reducing the number of illustrations, we have also shortened the accompanying text throughout for conciseness and removed a sub-section heading to simplify Section 3’s structure.

Reviewer #2: 

COMMENT: This is a good study that empowers local communities to engage in water security. The ability to triangulate the outcomes between the surveys and the participatory mapping was critical to demonstrating the effectiveness of their methods. 

RESPONSE: We thank the reviewer for these positive comments; no other response needed.

COMMENT: However, the overall article ends weakly. In particular, the last two sentences let the reader down. The next steps and recommendations for future research should be clearly articulated at the end of the conclusion section. Some of this is hinted at in the discussion, finish strongly.

RESPONSE: Having revisited the literature on water safety planning and participatory mapping as suggested by Reviewer 1, we agree that we should have drawn out more in our Discussion and Conclusions. With this in mind, we have introduced a new sixth paragraph in the Discussion, where we reflect on the complexity of rural water source use and the value of informal and household-managed sources to the communities. We now highlight the application of rural water safety planning to such sources as a specific area for future research via an extended Conclusion.

---

## [Decision Letter · Decision Letter 1]

17 Jun 2021

PONE-D-20-28371R1

A mixed methods study to evaluate participatory mapping for rural water safety planning in western Kenya

PLOS ONE

Dear Dr. Wright,

Thank you for submitting your manuscript to PLOS ONE. After careful consideration, we feel that it has merit but does not fully meet PLOS ONE’s publication criteria as it currently stands. Therefore, we invite you to submit a revised version of the manuscript that addresses the points raised during the review process.

I have now received comments on your revised manuscript. I agree with the reviewers that the revised paper is significantly improved. Reviewer 1 has minor concerns that you need to satisfactorily address before your manuscript is accepted for publication. 

We look forward to receiving your revised manuscript.

Kind regards,

Frank Onderi Masese, Ph.D

Academic Editor

PLOS ONE

Journal Requirements:

Additional Editor Comments (if provided):

The reviewers have gone through your responses and note that the paper is significantly improved. I invite you once more to make corrections and /or clarifications to comments by reviewer 1.

Reviewers' comments:

Reviewer's Responses to Questions

**Comments to the Author**

1. If the authors have adequately addressed your comments raised in a previous round of review and you feel that this manuscript is now acceptable for publication, you may indicate that here to bypass the “Comments to the Author” section, enter your conflict of interest statement in the “Confidential to Editor” section, and submit your "Accept" recommendation.

Reviewer #1: All comments have been addressed

Reviewer #2: All comments have been addressed

2. Is the manuscript technically sound, and do the data support the conclusions?

Reviewer #1: Yes

Reviewer #2: Yes

3. Has the statistical analysis been performed appropriately and rigorously? 

Reviewer #1: N/A

Reviewer #2: Yes

4. Have the authors made all data underlying the findings in their manuscript fully available?

Reviewer #1: Yes

Reviewer #2: Yes

5. Is the manuscript presented in an intelligible fashion and written in standard English?

Reviewer #1: Yes

Reviewer #2: Yes

6. Review Comments to the Author

Reviewer #1: just one or two minor revisions. see attached table.

Please see the attached table.

1) Confusing sentence: “perceptions of what constitutes a contamination hazard consequent implications for hazard management 75-76 Sentence was not changed.

2) What is the status of rural water safety planning generally speaking, in Africa, and in Kenya particularly? 62-64 Generally, Sub-Saharan Africa has lagged behind other world regions, with initial uptake in Uganda by 2011 (4), and a with subsequent more widespread uptake elsewhere in Africa including by Kenyan utilities (5). More information on the status of rural water safety planning in Kenya specifically would help strengthen this

3) Why is this method optimal? Incomplete

Reviewer #2: The changes made in this revision have greatly improved the usefulness of this work. The manuscript has been tightened up, but more importantly, the implications of the study have been made clear so that the reader can better understand the context of the study and the implications of the results.

7. PLOS authors have the option to publish the peer review history of their article (what does this mean?). If published, this will include your full peer review and any attached files.

Reviewer #1: No

Reviewer #2: No

---

## [Author Response · Author response to Decision Letter 1]

27 Jun 2021

We thank reviewer 1 for taking time to provide a very diligent and thorough review of our manuscript, which we also believe has greatly improved it. Responses to the remaining outstanding comments are provided below.

COMMENT 1) Confusing sentence: “perceptions of what constitutes a contamination hazard consequent implications for hazard management 75-76 Sentence was not changed.

RESPONSE: We have reworded this sentence to make our meaning clearer (paragraph 2 of Introduction).

COMMENT: 2) What is the status of rural water safety planning generally speaking, in Africa, and in Kenya particularly? 62-64 Generally, Sub-Saharan Africa has lagged behind other world regions, with initial uptake in Uganda by 2011 (4), and a with subsequent more widespread uptake elsewhere in Africa including by Kenyan utilities (5). More information on the status of rural water safety planning in Kenya specifically would help strengthen this.

RESPONSE: One of the reasons why we struggled to respond to this component of the reviewer’s suggestion in our previous submission is that there is actually minimal published evidence on rural water safety planning in Kenya. However, in our revised version, we now summarise how far the latest government guidance in Kenyan covers the situation in rural areas (final sentences of paragraph 3 in our revised Introduction).

COMMENT: 3) Why is this method optimal? Incomplete

RESPONSE: We now briefly summarise the reasons as to why participatory mapping is very well suited to water safety planning, drawing on WHO recommendations advocating participatory mapping for water safety planning of small community supplies in doing so (Introduction, penultimate paragraph).

---

## [Editor Report · Decision Letter 2]

14 Jul 2021

A mixed methods study to evaluate participatory mapping for rural water safety planning in western Kenya

PONE-D-20-28371R2

Dear Dr. Wright,

We’re pleased to inform you that your manuscript has been judged scientifically suitable for publication and will be formally accepted for publication once it meets all outstanding technical requirements.

Kind regards,

Frank Onderi Masese, Ph.D

Academic Editor

PLOS ONE
---

## [Editor Report · Acceptance letter]

19 Jul 2021

PONE-D-20-28371R2 

A mixed methods study to evaluate participatory mapping for rural water safety planning in western Kenya 

Dear Dr. Wright:

I'm pleased to inform you that your manuscript has been deemed suitable for publication in PLOS ONE. Congratulations! Your manuscript is now with our production department. 

Kind regards, 

on behalf of

Dr. Frank Onderi Masese 

Academic Editor

PLOS ONE